# FIDELITY-BASED DEEP ADIABATIC SCHEDULING

**Eli Ovits & Lior Wolf**
Tel Aviv University

## ABSTRACT

Adiabatic quantum computation is a form of computation that acts by slowly interpolating a quantum system between an easy to prepare initial state and a final state that represents a solution to a given computational problem. The choice of the interpolation schedule is critical to the performance: if at a certain time point, the evolution is too rapid, the system has a high probability to transfer to a higher energy state, which does not represent a solution to the problem. On the other hand, an evolution that is too slow leads to a loss of computation time and increases the probability of failure due to decoherence. In this work, we train deep neural models to produce optimal schedules that are conditioned on the problem at hand. We consider two types of problem representation: the Hamiltonian form, and the Quadratic Unconstrained Binary Optimization (QUBO) form. A novel loss function that scores schedules according to their approximated success probability is introduced. We benchmark our approach on random QUBO problems, Grover search, 3-SAT, and MAX-CUT problems and show that our approach outperforms, by a sizable margin, the linear schedules as well as alternative approaches that were very recently proposed.

## 1 INTRODUCTION

Many of the algorithms developed for quantum computing employ the quantum circuit model, in which a quantum state involving multiple qubits undergoes a series of invertible transformations. However, an alternative model, called Adiabatic Quantum Computation (AQC) (Farhi et al., 2000; McGeoch, 2014), is used in some of the leading quantum computers, such as those manufactured by D-Wave Systems (Boixo et al., 2014). AQC algorithms can achieve quantum speedups over classical algorithms (Albash & Lidar, 2018), and are polynomially equivalent to the quantum circuit model (Aharonov et al., 2008).

In AQC, given a computational problem $Q$, e.g., a specific instance of a 3SAT problem, a physical system is slowly evolved until a specific quantum state that represents a proper solution is achieved. Each AQC run involves three components:

1. An initial Hamiltonian $\mathcal{H}_b$, chosen such that its ground state (in matrix terms, the minimal eigenvector of $\mathcal{H}_b$) is easy to prepare and there is a large spectral gap. This is typically independent of the specific instance of $Q$.

2. A final Hamiltonian $\mathcal{H}_p$ designed such that its ground state corresponds to the solution of the problem instance $Q$.

3. An adiabatic schedule, which is a strictly increasing function $s(t)$ that maps a point in time $0 \leq t \leq t_f$, where $t_f$ is total computation time, to the entire interval $[0, 1]$ (i.e., $s(0) = 0$, $s(t_f) = 1$, and $s(t_1) < s(t_2)$ iff $t_1 < t_2$ and vice versa).

These three components define a single time-dependent Hamiltonian $\mathcal{H}(t)$, which can be seen as an algorithm for solving $Q$:

$$\mathcal{H}(t) = (1 - s(t)) \cdot \mathcal{H}_b + s(t) \cdot \mathcal{H}_p \tag{1}$$

At the end of the adiabatic calculation, the quantum state is measured. The square of the overlap between the quantum state and ground state of the final Hamiltonian, is the fidelity, and represents the probability of success in finding the correct solution. An AQC algorithm that is evolved over an insufficient time period (a schedule that is too fast) will have a low fidelity. Finding the optimal

schedule, i.e., the one that would lead to a high fidelity and would keep the time complexity of the algorithm minimal is, therefore, of a great value. However, for most problems, an analytical solution for the optimal schedule does not exist (Albash & Lidar, 2018).

Attempts were made to optimize specific aspects of the adiabatic schedule by using iterative methods (Zeng et al., 2015) or by direct derivations (Susa et al., 2018). Performance was evaluated by examining characteristics of the resulting dynamic (e.g. the minimum energy gap) and no improvement was demonstrated on the full quantum calculation.

Previous attempts to employ AI for the task of finding the optimal schedule have relied on Reinforcement Learning (Lin et al., 2020; Chen et al., 2020). While these methods were able to find schedules that are better than the linear path, they are limited to either learning one path for a family of problems (without considering the specific instance) or to rerunning the AQC of a specific instance $Q$ multiple times in order to optimize the schedule.

In our work, supervised learning is employed in order to generalize from a training set of problems and their optimal paths to new problem instances. Training is done offline and the schedule our neural model outputs is a function of the specific problem instance. The problem instance is encoded in our model either based on the final Hamiltonian $\mathcal{H}_p$ or directly based on the problem.

The suggested neural models are tested using several different problem types: Grover search problems, 3SAT and MAX-CUT problems, and randomized QUBO problems. We show that the evolution schedules suggested by our model greatly outperform the naive linear evolution schedule, as well as those schedules provided by the recent RL methods, and allow for much shorter total evolution times.

## 2 BACKGROUND

The goal of the scheduling task is to find a schedule $s(t)$ that maximizes the probability to get the correct answer for instance $Q$, using $H_b$ and $H_p$ over an adiabatic quantum computer. The solution to $Q$ is coded as the lowest energy eigenstate of $\mathcal{H}_p$.

In order to achieve the solution state with high probability, the system must be evolved "sufficiently slowly". The adiabatic theorem (Roland & Cerf, 2002; Albash & Lidar, 2018; Rezakhani et al., 2009) is used to analyze how fast could this evolution be. It states that the probability to reach the desired state at the end of the adiabatic calculation is $1 - \varepsilon^2$ for $\varepsilon << 1$ if

$$\frac{\left|\langle E_1(t)| \frac{d}{dt}\mathcal{H}(t) |E_0(t)\rangle\right|}{g^2(t)} \leq \varepsilon \tag{2}$$

where the Dirac notation (Tumulka, 2009) is used[1], $E_0(t)$ ($E_1(t)$) is the ground state (first excited state) of the time dependent Hamiltonian $\mathcal{H}(t)$, i.e., the eigenstates that corresponds to the lowest (2nd lowest) eigenvalue, and $g(t)$ is the time dependent instantaneous spectral gap between the smallest and second smallest eigenvalues of $\mathcal{H}(t)$.

Let $t_f$ be the total calculation time. let $s(t)$ be an evolution schedule, such that $s(0) = 0$, $s(t_f) = 1$. Applying the adiabatic condition for $s(t)$, we get

$$\frac{\left|\langle E_1(s(t))| \frac{ds}{dt}\frac{d}{ds}\mathcal{H}(s(t)) |E_0(s(t))\rangle\right|}{g^2(s(t))} \leq \varepsilon \Rightarrow \frac{ds}{dt} \leq \varepsilon \frac{g^2(s)}{\left|\langle E_1(s)| \frac{d}{ds}\mathcal{H}(s) |E_0(s)\rangle\right|} \tag{3}$$

we could solve for $t(s)$ by integration to get

$$t(s) = \frac{1}{\varepsilon} \int_0^s \frac{\left|\langle E_1(s)| \frac{d}{ds}\mathcal{H}(s) |E_0(s)\rangle\right|}{g^2(s)} ds \tag{4}$$

and the total required evolution time is

$$t_f = t(s=1) = \frac{1}{\varepsilon} \int_0^1 \frac{\left|\langle E_1(s)| \frac{d}{ds}\mathcal{H}(s) |E_0(s)\rangle\right|}{g^2(s)} ds \tag{5}$$

---

[1]See appendix A for the conventional matrix notation.

We note that finding a numerical solution for eq 4 requires calculating the full eigenvalue decomposition of $\mathcal{H}(x)$.

## 2.1 MOST-RELATED WORK

Two recent contributions use deep learning in order to obtain, for a given $t_f$, a schedule that outperform the linear schedule. Lin et al. (2020) suggest using deep reinforcement learning in order to find an optimal schedule for each specific class of problems (e.g., 3SAT problems of a certain size). In contrast, we study the problem of finding schedules for generic problem instances. They train and benchmark their performance by simulating an adiabatic quantum computer, and scoring the computation results for randomly chosen problem instances. Their results are generally better than the naive linear schedule, and the solution produced by their neural network is somewhat transferable for larger problem sizes.

Chen et al. (2020) also use RL to construct, given a $t_f$, a schedule for 3SAT problems. The most successful technique suggested is a Monte Carlo Tree Search (MCTS, Silver et al. (2016)), which produces results that significantly outperform the linear schedule. This technique requires running the adiabatic evolution process many times for each problem, in order to find a successful schedule. An approach inspired by alpha-zero (Silver et al., 2018) is used to adapt the generic MCTS solution to specific problem class, while requiring only a few additional rounds of the adiabatic process for each new instance. In our method, we do not require any run given a new problem instance.

## 3 METHOD

We consider two types of deep neural models. The first model is designed to get the problem Hamiltonian $\mathcal{H}_p$ as an input. For an $n$ qubit problem, the problem Hamiltonian is generally of size $2^n \times 2^n$. In this work, we consider problem Hamiltonians which are diagonal and can be represented by vector of size $2^n$. This scenario covers both the Grover search problem and the 3SAT problem we present in Sec. 4.

The second model is designed to get a quadratic unconstrained binary optimization (QUBO) problem as an input. The QUBO problem has the following form:

$$\bar{x} = argmin_x(x^T Q x), \tag{6}$$

where $x$ is a vector of binary variables and $Q \in \mathbb{R}^{n \times n}$ defines the specific QUBO instance. The QUBO problem is NP-Complete, and many types of common problems can be reduced to QUBO (Glover et al., 2018). The QUBO formulation is of special interest in the context of adiabatic quantum computing, since it allows a relatively easy mapping to real quantum annealing devices that do not possess full qubit connectivity (Cruz-Santos et al., 2019).

A QUBO problem Q can be converted to the Hamiltonian form in the following fashion:

$$\mathcal{H}_p = \sum_{i=1}^{n} Q_{ii}\left(\frac{I + \sigma_z^i}{2}\right) + \sum_{i \neq j} Q_{ij}\left(\frac{I + \sigma_z^i}{2}\right)\left(\frac{I + \sigma_z^j}{2}\right), \tag{7}$$

where $\sigma_z^i$ is the Pauli matrix $\sigma_z$ operating only on qubit i (Liboff, 2003). The resulting $\mathcal{H}_p$ is of size $2^n \times 2^n$ and is diagonal.

The prediction target of our models is the desired normalized schedule $\hat{s}(t)$, which is defined over the range $[0, 1]$ as $\hat{s}(t) = s(t/t_f)$. For the purpose of estimation, it is sampled at 100 points in the interval $t = [0, 1]$. The representation of this schedule is given as a vector $\boldsymbol{d} \in [0, 1]^{99}$, which captures the temporal derivative of the schedule. In other words, $\boldsymbol{d}$ is trained to hold the differences between consecutive points on the path, i.e., element $i$ is given by $\boldsymbol{d}_i = \hat{s}((i+1)/100) - \hat{s}(i/100)$. Note that the sum of $\boldsymbol{d}$ is one.

## 3.1 UNIVERSALITY OF THE OPTIMAL SCHEDULE

The reason that we work with the normalized schedule is that the optimal evolution schedule is not dependent upon the choice of $t_f$. As shown next, for every time budget $t_f$, the same normalized schedule would provide the highest fidelity (neglecting decoherence).

Let $s_1(t) : [0, t_f] \to [0, 1]$ be a suggested evolution schedule, which outperforms a different suggested schedule $s_2(t)$, for a specific $t_f = \tau_1$, i.e. it achieves a greater fidelity at the end of the schedule for a specific problem instance Q. Then, Thm. 1 shows that $s_1(t)$ outperforms $s_2(t)$ for every possible choice of $t_f$ for the same problem Q.

**Theorem 1.** *Let $s_1(t)$ and $s_2(t)$ be two monotonically increasing fully differentiable bijective functions from $[0, t_f = \tau_1]$ to $[0, 1]$. Let Q be an optimization problem, and assume that $s_1(t)$ achieves a greater fidelity than $s_2(t)$ at the end of a quantum adiabatic computation for Q with total evolution time $t_f = \tau_1$. Then, for any other choice $t_f = \tau_2$, the scaled schedule $s_1(\frac{\tau_2}{\tau_1}t)$ will achieve a greater fidelity than $s_2(\frac{\tau_2}{\tau_1}t)$ for an adiabatic computation over the same problem Q with total evolution time $t_f = \tau_2$.*

The proof can be found in appendix B.

## 3.2 ARCHITECTURE

The model architectures are straightforward and no substantial effort was done to optimize them. The Hamiltonian as input model has seven fully connected layers, with decreasing sizes: 4096, 2048, 2048, 1024, 512, and finally the output layer, which, as mentioned, is of size 99.

For the QUBO model, in which the input is a matrix, a two part architecture was used. In the first part, five layers of 2D convolution was employed, with kernel size of $3 \times 3$, for 64 kernels. The output from the convolution layers was then flattened to a vector of size $64n^2$, and fed to the second part of the network, consisted of five fully connected layers, with decreasing dimensions of 2048, 1024, 1024, 512, and finally the output layer of size 99.

This output layers in both models are normalized to have a sum of one. For both models, the SELU activation function Klambauer et al. (2017) was used for all layers, except the final layer, which used the sigmoid (logistic) function.

## 3.3 A FIDELITY BASED LOSS FUNCTION

Let $|\psi(t)\rangle$ is the state of the quantum system at time $t = st_f$. The fidelity of the QAC is given by (Farhi et al., 2000)

$$p_{success} = |\langle E_0(s = 1) | \psi(t = t_f)\rangle|^2 , \tag{8}$$

where $\langle E_\ell(s = 1)|$ is the $\ell$-th eigenstate of the parameter dependent evolution Hamiltonian $\mathcal{H}(s)$, such that $\langle E_0(s = 1)|$ is the ground state of the final Hamiltonian $\mathcal{H}_p$. Finding $\langle E_0(s = 1)|$ requires performing eigenvalue decomposition for $\mathcal{H}_p$, which is equivalent to solving the original optimization problem, and is done for the training set.

The quantum state $|\psi(t)\rangle$ is evolving according to the Schrödinger equation

$$i\frac{d}{dt} |\psi(t)\rangle = \mathcal{H}(t) |\psi(t)\rangle \tag{9}$$

A brute force approach for finding $p_{success}$ is to numerically solve the Schrödinger equation, see appendix C. This full numerical calculation is, however, too intense to be practical. We next develop an approximate method that would be easier to compute and still be physically meaningful. It is based on the adiabatic local evolution speed limit from Eq. 3:

$$\left|\frac{ds}{dt}\right| \le \varepsilon \frac{g^2(s)}{\left|\langle E_1(s)| \frac{d}{ds}\mathcal{H}(s) |E_0(s)\rangle\right|} \tag{10}$$

This inequality could be used as a local condition for convergence of any suggested path. We define

$$g_E^2(s) = \frac{g^2(s)}{\left|\langle E_1(s)| \frac{d}{ds}\mathcal{H}(s) |E_0(s)\rangle\right|} \tag{11}$$

We would like to use the local condition to create a global convergence condition for a full suggested path $s(t), 0 \le t \le t_f$. To do so, we integrate both sides of Eq. 10 over the suggested schedule $s$.

This integral represents a mean value of the local adiabatic condition, for every point in the suggested schedule.

$$\varepsilon = \int\limits_0^1 \frac{\frac{ds}{dt}}{g_E^2(s)} ds \tag{12}$$

We note that integrand is always positive (assuming $s(t)$ is monotonically increasing). Recall that the adiabatic theorem ties $\varepsilon$ to the fidelity: $\varepsilon = \sqrt{1 - p_{success}}$. By defining the right hand side of Eq.12 as our loss function, we ensure that any training process that minimizes Eq. 12 will maximize the fidelity. Recall that the vector $d$ that the network outputs is a vector of differences, therefore, it approximates the local derivatives of the obtained path. Let $\hat{s}^*$ be the optimal normalized path, which we estimate for each training sample. The loss function is, therefore, defined as:

$$\mathcal{L}(d, \hat{s}^*) = \sum_{i=1}^{99} \frac{d_i^2}{g_E^2(\hat{s}^*(i/100))} \tag{13}$$

The values of $g_E$ are precomputed along the optimal path $\hat{s}^*$ for efficiency. While the denominator is obtained on points that do not correspond to the estimated path (the commutative sum of $d$), the approximation becomes increasingly accurate at the estimated path appraoches the optimal one.

### 3.4 TRAINING DATA AND THE TRAINING PROCESS

In order to train the QUBO problem model, we produced a training dataset of 10,000 random QUBO instances for each problem size: $n = 6, 8, 10$. The QUBO problems were generated by sampling independently, from the normal distribution, each coefficient of the problem matrix $Q$. The entire matrix $Q$ was then multiplied by a single random normal variable.

We approximated an optimal evolution schedule for each problem, by calculating the full eigenvalue decomposition of $\mathcal{H}_t$ as described in Sec 2. We also calculated the value of $g(s(t))$ for each problem.

For the model that uses the problem Hamiltonian as input, we used the same prepared QUBO problems, converted to the Hamiltonian form. In addition, we added another 500 cases of randomized Hamiltonians with randomized values around distinct energy levels. For each Hamiltonian, We first randomized an energy level between the following values: 0.5, 1, 1.5 or 2, and then randomized uniformly distributed values around the selected energy level. To each Hamiltonian we added a single ground state with energy 0. This type of Hamiltonian is not commonly created by the random QUBO creation process described above, but is more representative of binary optimization problems, and specifically more closely resembles problem Hamiltonians for the Grover problem and the 3SAT problem, which we later use to benchmark our model performance. We note that the Hamiltonian for these specific problems in our test set are nevertheless different from our randomized problem Hamiltonians, which highlights the generalization capability of our method.

The training was performed using the Adam optimizer (Kingma & Ba, 2014), with batches of size 200. Batch normalization (Ioffe & Szegedy, 2015) was applied during training. A uniform dropout value of 0.1 is employed for all layers during the model training.

## 4 RESULTS

As a baseline to the loss $\mathcal{L}$ (Eq. 13) we use, we employed the Mean Squared Error (MSE) loss, for which the model output was compared to the known optimal schedule from the dataset, which was calculated in advance.

### 4.1 GROVER SEARCH

The Grover algorithm is a well-known quantum algorithm that finds with high probability the unique input to a black box function that produces a particular output value, using just $\sqrt{N}$ evaluations of the function, where $N$ is size of the search space. For an $n$ qubit space, the search is over the set $\{0, 1, .., 2^n - 1\}$, making $N = 2^n$. It is possible to reproduce the Grover speedup using an adiabatic formulation, with the following problem Hamiltonian:

$$\mathcal{H}_p = I - |m\rangle \langle m| , \tag{14}$$

where $|m\rangle$ is the state that represents the value we search. Roland & Cerf (2002) showed that for this problem, a linear schedule does not produce quantum speedup over a classical algorithm, but for a specific initial Hamiltonian $\mathcal{H}_b = I - |\psi_0\rangle \langle\psi_0|$, for $\psi_0$ as the maximal superposition state (a sum of the states representing all values from 0 to $N - 1$), an optimal schedule could be derived analytically to achieve a quadratic speedup. The optimal path is given by

$$\hat{s}(t) = \frac{1}{2} + \frac{1}{2\sqrt{N-1}} \tan\left[(2s-1)\tan^{-1}\sqrt{N-1}\right] \tag{15}$$

In practice, the proposed $\mathcal{H}_b$ is hard to physically realize, and a simpler initial Hamiltonian is used:

$$\mathcal{H}_b = \frac{1}{2} \sum_{i=1}^{n} I - \sigma_x^i, \tag{16}$$

where $\sigma_x^i$ is the Pauli matrix $\sigma_x$ operating only on qubit i (Liboff, 2003).

We test our model's performance by using the Grover problem Hamiltonian $\mathcal{H}_p$ as input for several problem sizes. Different Grover problems are completely symmetrical, and are identical after changing variables, so it is sufficient to use a single test case to test our model.

We benchmark our model's performance by simulating AQC for multiple values of $t_f$, and calculating the fidelity by measuring the overlap between the quantum state at the end of the adiabatic evolution and the solution state.

We also show the convergence pattern for the fidelity (i.e. the overlap with the solution state, measured during the adiabatic evolution) for a single specific $t_f$. For each problem size, we chose a different $t_f$, for which a full convergence ($p > 0.95$) is achieved with the evolution schedule suggested by our model. We compare several suggested schedules: the path produced by training our model using our novel loss function, the path produced by training our model using the MSE loss, the linear path, and a numerically calculated optimal path. We also include the results reported by Lin et al. (2020) for the same problem.

The results are reported in Fig. 1 for $n = 6, 10$, see appendix for $n = 8$. It is evident that our model produces paths that are significantly superior to the linear path, and also outperforms Lin et al. (2020). The advantage of the new loss function over the MSE loss is also clear.

Recall that for a Grover search with a certain $n$, $\mathcal{H}_p$ is a diagonal matrix of size $2^n \times 2^n$. To check whether the model trained on $n = 10$ generalizes to larger search problems, we view the diagonal of $\mathcal{H}_p$ for $n' > n$ as a 1D signal. This signal is smoothed by a uniform averaging mask of size $6\frac{2^{n'}}{2^n}$, and subsampled to obtain a diagonal of size $2^n$.

The results are presented in Fig. 2. Evidently, the network trained for $n = 10$ achieves much better results than the linear baseline for sizes $n' = 12, 14, 16$. We also trained a network for $n = 16$. As can be seen in Fig. 2(c), this network does achieve better fidelity than the smaller network. We note that no significant changes were made to the network architecture, and the only difference is in the size of the input layer. Appendix D presents results for the $n = 16$ network on $n' = 17, .., 20$. Our $\mathcal{L}$-trained model achieves a much better fidelity than the linear schedule and the MSE baseline.

## 4.2 3SAT

In the 3-SAT problem, the logical statement consists of $m$ clauses, $C_i$, such that each clause contain a disjunction over three variables out of $n$ binary variables. A solution to the 3SAT problem is an assignment for the $n$ variables that satisfies all $m$ clauses. It is possible to construct a problem Hamiltonian for each 3SAT problem, by taking a sum over all clauses

$$\mathcal{H}_p = \frac{1}{2} \sum_{i=1}^{m} I + \sigma_z^{F_i}, \tag{17}$$

where $\sigma_z^{F_i}$ is the Pauli matrix $\sigma_z$ operating only on the state that represents the assignment $|a = \{0, 1\}, b = \{0, 1\}, c = \{0, 1\}\rangle$ which produces False value for clause i. This Hamiltonian counts the number of clauses which are not satisfied by each assignment, and its ground state corresponds to the eigenvalue 0 and represents the solution of the problem, for which all clauses are satisfied.

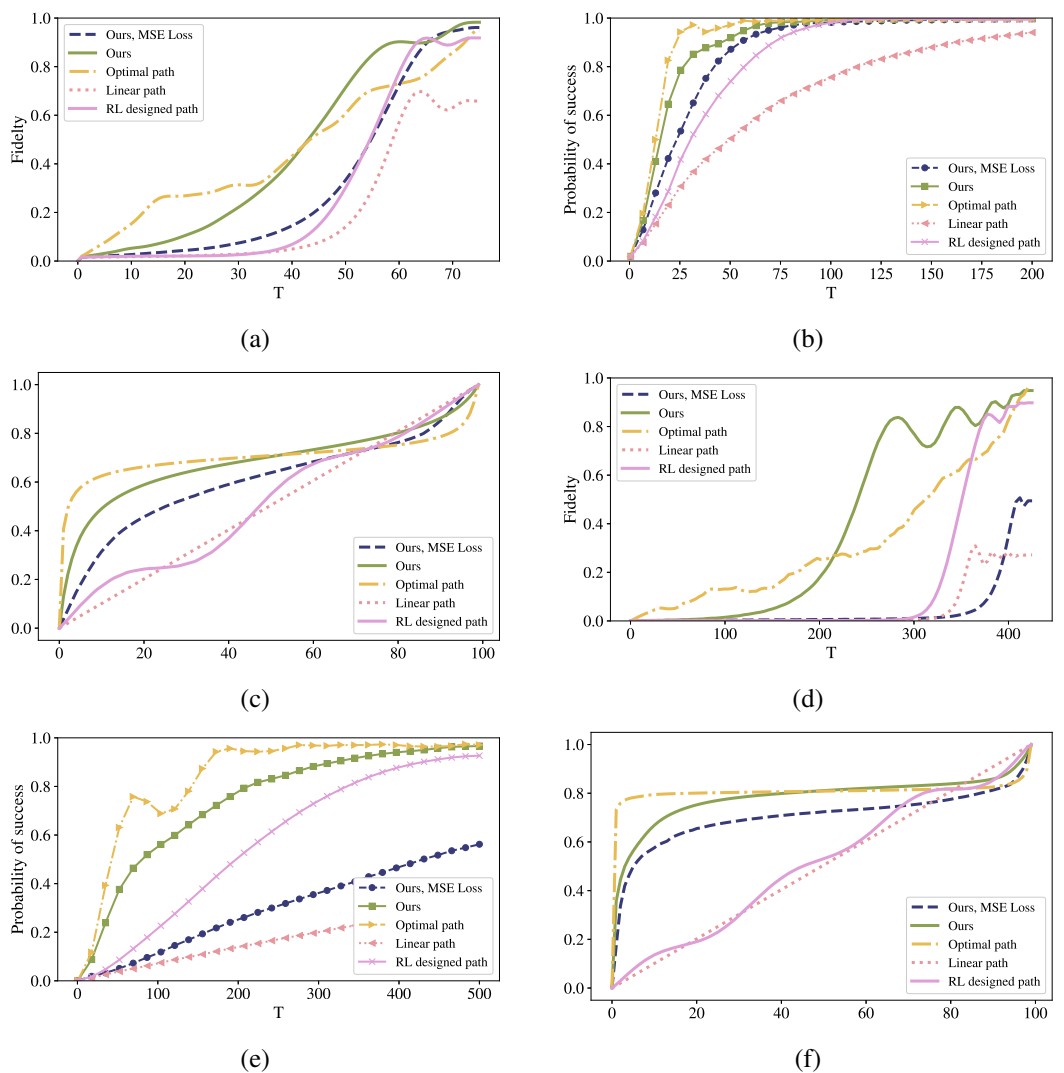

Figure 1: Results for Grover search over (a-c) n=6 or (d-f) n=10 qubits. (a) fidelity for $t_f = 75$, and (d) fidelity for $t_f = 425$. (b,e) fidelity at time $t_f$ for multiple $t_f$ values. (c,f) suggested schedules.

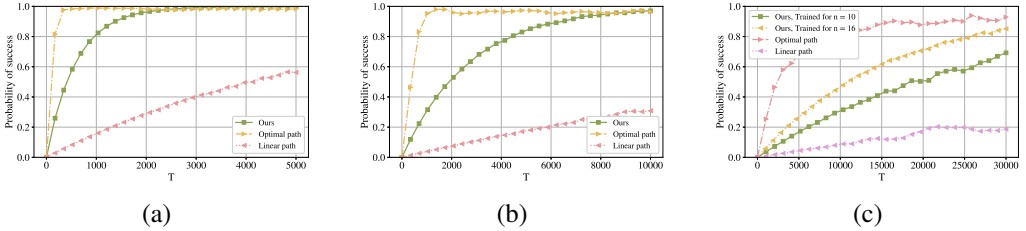

Figure 2: Fidelity for various $t_f$ values for Grover search problems of size $n'$ larger than the $n = 10$, for which the network was trained. (a) $n' = 12$, (b) $n' = 14$, (c) $n' = 16$. For $n' = 16$, we also present the result obtained by a network trained for solving $n = 16$.

We test our model's performance, by randomizing 3SAT problems, and converting them to Hamiltonian form. Following Chen et al. (2020), we focus on 3SAT problems with a single solution, and a number of clauses $m = 3n$. This type of 3SAT problems is considered difficult to solve with adiabatic algorithms (Žnidarič, 2005).

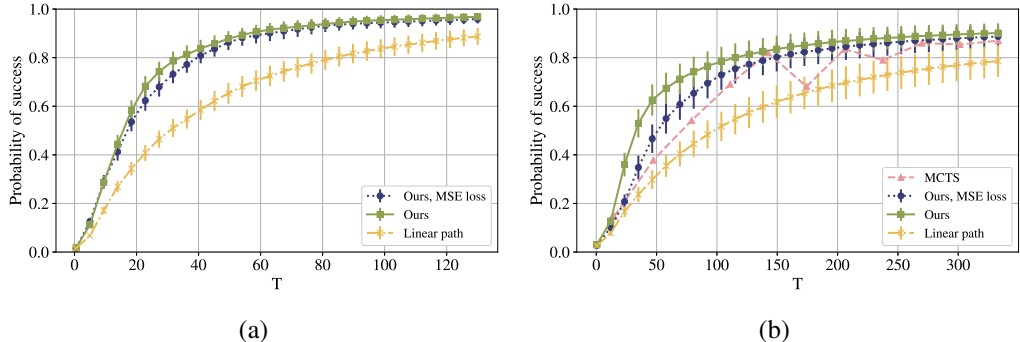

(a)                                        (b)

Figure 3: Fidelity for various $t_f$ values, over random 3SAT instances with m=3n clauses. Keep note of the different time scale for each problem size, as larger problems require longer times to converge. (a) n=8 qubits (b) n=11 qubits. For $n = 11$, we employ the test set of Chen et al. (2020) and directly compare with their MCTS method.

We benchmark our model's performance by simulating the adiabatic computation for multiple values of $t_f$ and calculating the fidelity by measuring the overlap between the quantum state at the end of the adiabatic evolution and the solution state.

In addition to the linear path and the paths obtained by training with either $\mathcal{L}$ or MSE, we also include for n=11, the results for the schedules designed by MCTS (Chen et al., 2020). For this purpose, we used the test data obtained by Chen et al. As can be seen in Fig. 3, our method outperform all baselines. Note that the MCTS methdod was optimized, for each problem instance and for each $t_f$ using tens AQC of runs on the specific test problem, while our method does not run on the test data.

As stated in Sec. 3.4, the Hamiltonian model is trained on 10,000 random QUBO problems and 500 random Hamiltonian problems. In Appendix E, we study the performance when the 500 random samples are removed from the training set and when employing fewer training samples.

## 4.3  MAX-CUT

To further demonstrate the generalization capability of the trained model, our Hamiltonian model for size n=10 is tested on random MAX-CUT problems. In a graph, a maximum cut is a partition of the graph's vertices into two complementary sets, such that the total edges weight between the sets is as large as possible. Finding the maximum cut for a general graph is known to be an NP-complete problem (MAX-CUT).

To generate random MAX-CUT problem instances, we choose a random subset of edges that contains at least half of the edges of the fully connected graph. We then sample the weights of each edge uniformly. When converting a MAX-CUT problem to the Hamiltonian form, $n$ is the number of vertices in the graph (Goto et al., 2019).

Fig. 4 presents the results of our our method for both with $\mathcal{L}$ and MSE, as well as the linear path. The results were averaged over 50 runs and conducted for $n = 10$. As can be seen, our complete method outperforms the baselines.

## 4.4  QUBO

To test our models with general QUBO problems, sets of random QUBO test problems of varying difficulty are generated . Since the final energy gap of the corresponding problem Hamiltonian is a critical parameter that determines the difficulty of the problem at hand (problems with a small energy gap require much longer evolution schedules), we generated two sets of test problems. The first has an energy gap of $g \sim 10$ and the second has an energy gap of $g \sim 0.1$. Varying the gap was obtained by multiplying the random Q matrix by the required values of the gap.

We benchmark the model's performance as in previous problems. However, in this case, we have two alternatives models: the one the receives the matrix $Q$ as input and the one that receives the Hamiltonian $\mathcal{H}_p$.

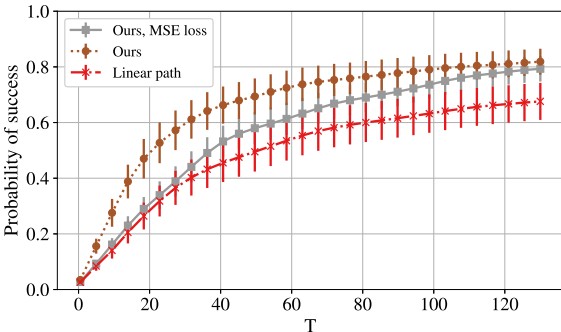

Figure 4: Fidelity for various $t_f$ values, for random $n = 10$ MAX-CUT instances.

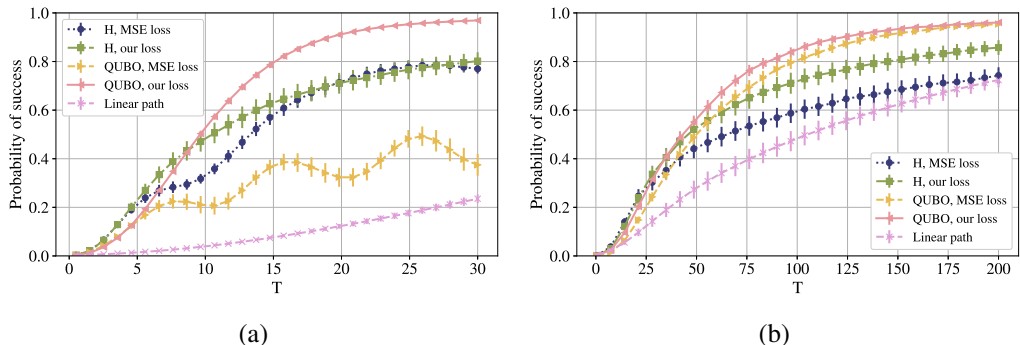

Figure 5: Fidelity for various $t_f$ values, over random QUBO instances of size n=10. Keep note of the different time scale for each problem, as problems with different energy gaps require different times to converge. The energy gaps are (a) $g \sim 10$, (b) $g \sim 0.1$

We noticed that the process of creating samples of varying spectral gaps creates a mismatch in scale with the training set problems. To compensate, we pre-process the inputs to the network models. Specifically, for the model that has $Q$ as input, we normalize the Frobenius norm of each $Q$ such that if it is larger than 60, we scale $Q$ to have a norm of 60. Similarly for the model that accepts $\mathcal{H}_p$ as input, we clip every value that is larger than 90 to be 90 ($Q$ with high norms translate to Hamiltonians with specific coeefieicents that are high). To clarify, this preprocessing is only applied to the input to the models and does not change the problem we solve.

Our Results are presented at Fig. 5. As can be seen, our dedicated QUBO model (Q as input) constructs successful schedules, outperforming all other models. The Hamiltonian model trained with our loss obtains the second highest results. The advantage of the fidelity-based loss term is evident in all cases.

For a further comparison between the $\mathcal{L}$ loss term and MSE, please refer to Appendix F.

## 5 CONCLUSIONS

Optimal scheduling of AQC tasks is the main way to reduce the time complexity for an emerging class of quantum computes. While recent work has applied RL for this task, it either provided a generic schedule for each class of problems or required running the exact computation that needs to be solved multiple times. Our solution employs a separate training set, and at test time provides a schedule that is tailored to the specific instance, without performing any runs. Remarkably, although our training was performed for one type of problem (QUBO), it generalizes well to completely different instances: Grover search, 3-SAT, and MAX-CUT. At the heart of our method lies a new type of loss that maximizes the fidelity based on a new approximation of the success probability. Our experiments demonstrate the effectiveness of our method, as well as its advantage over the recent contributions.

ACKNOWLEDGEMENTS

This project has received funding from the European Research Council (ERC) under the European Union's Horizon 2020 research and innovation programme (grant ERC CoG 725974).

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

## A  CONVENTIONAL MATRIX NOTATION

For those not familiar with the Dirac notation, we repeat the equations in conventional matrix multiplication notation.

$$\frac{\left|E_1(t)^\top (\frac{d}{dt}\mathcal{H})E_0(t)\right|}{g^2(t)} \leq \varepsilon \tag{2}$$

$$\frac{\left|E_1(s(t))^\top \frac{ds}{dt}\frac{d}{ds}\mathcal{H}(s(t))E_0(s(t))\right|}{g^2(s(t))} \leq \varepsilon \Rightarrow \left|\frac{ds}{dt}\right| \leq \varepsilon\frac{g^2(s)}{\left|E_1(s)^\top \frac{d}{ds}\mathcal{H}(s)E_0(s)\right|} \tag{3}$$

$$t(s) = \frac{1}{\varepsilon}\int_0^s \frac{\left|E_1(s)^\top \frac{d}{ds}\mathcal{H}(s)E_0(s)\right|}{g^2(s)}ds \tag{4}$$

$$t_f = t(s=1) = \frac{1}{\varepsilon}\int_0^1 \frac{\left|E_1(s)^\top \frac{d}{ds}\mathcal{H}(s)E_0(s)\right|}{g^2(s)}ds \tag{5}$$

$$p_{success} = \left|E_0(s=1)^\top \psi(t=t_f)\right|^2 \tag{8}$$

$$i\frac{d}{dt}\psi(t) = \mathcal{H}(t)\psi(t) \tag{9}$$

$$\left|\frac{ds}{dt}\right| \leq \varepsilon\frac{g^2(s)}{\left|E_1(s)^\top \frac{d}{ds}\mathcal{H}(s)E_0(s)\right|} \tag{10}$$

$$g_E^2(s) = \frac{g^2(s)}{\left|E_1(s)^\top \frac{d}{ds}\mathcal{H}(s)E_0(s)\right|} \tag{11}$$

$$\mathcal{H}_p = I - m^T m \tag{14}$$

## B  PROOF OF THM. 1

**Theorem 1.** *Let $s_1(t)$ and $s_2(t)$ be two monotonically increasing fully differentiable bijective functions from $[0, t_f = \tau_1]$ to $[0, 1]$. Let Q be an optimization problem, and assume that $s_1(t)$ achieves a greater fidelity than $s_2(t)$ at the end of a quantum adiabatic computation for Q with total evolution time $t_f = \tau_1$. Then, for any other choice $t_f = \tau_2$, the scaled schedule $s_1(\frac{\tau_2}{\tau_1}t)$ will achieve a greater fidelity than $s_2(\frac{\tau_2}{\tau_1}t)$ for an adiabatic computation over the same problem Q with total evolution time $t_f = \tau_2$.*

*Proof.* The adiabatic condition from Eq. 3 defines a local speed limit over the evolution schedule. We define:

$$g_E^2(s) = \frac{g^2(s)}{\left|\langle E_1(s)| \frac{d}{ds}\mathcal{H}(s) |E_0(s)\rangle\right|} \tag{18}$$

Then, for both schedules $s_i(t)$, $i = 1, 2$ the local adiabatic speed is

$$\frac{1}{g_E^2(s_i(t))} \frac{ds_i(t)}{dt} = \varepsilon_i(t), \ \ 0 \le t \le \tau_1 \tag{19}$$

We now consider a new $t_f = \tau_2$. We use the same suggested schedules with a scaling factor $a = \frac{\tau_1}{\tau_2}$:

$$s_i^{scaled}(t) = s_i(at) \tag{20}$$

It is clear that $s_1^{scaled}(t = 0) = 0$ and $s_1^{scaled}(t = \tau_2) = s_1\left(\frac{\tau_1}{\tau_2}\tau_2\right) = s_1(t = \tau_1) = 1$, and the same is true for $s_2^{scaled}(t)$. We calculate the new derivative

$$\frac{ds_i^{scaled}(t)}{dt} = a\frac{ds_i(at)}{dt} \tag{21}$$

By multiplying Eq. 19 by factor $a$ we can get for the new time axis $0 \le t \le \tau_2$

$$a\frac{1}{g_E^2(s_i(at))} \frac{ds_i(at)}{dt} = a \cdot \varepsilon_i(t) \tag{22}$$

then, we can switch to the scaled schedules and finally

$$\frac{1}{g_E^2(s_i^{scaled}(t))} \frac{ds_i^{scaled}(t)}{dt} = a \cdot \varepsilon_i(t) = \varepsilon_i^{scaled}(t) \tag{23}$$

We now consider the fidelity for each evolution schedule. According to the adiabatic theorem, the fidelity achieved at the end of the adiabatic evolution for each schedule is dependent solely on the local adiabatic speeds $\varepsilon_i(t)$. The resulting fidelity for the full path is then bounded by some functional $\mathcal{F} : L^2 \to \mathbb{R}$ which transforms all of the local adiabatic speeds to a single number.

$$p_i \ge 1 - \mathcal{F}\left(\varepsilon_i^2(t)\right) \tag{24}$$

Following Roland & Cerf (2002), we assume a global maximum value

$$\mathcal{F}[f(t)] = max(f(t)) \tag{25}$$

$$p_i \ge 1 - max\left(\varepsilon_i^2(t)\right) \tag{26}$$

it is clear that for this choice of $\mathcal{F}$,

$$\mathcal{F}[af(t)] = a\mathcal{F}[f(t)] \tag{27}$$

For any positive scalar $a$. It follows that the new values for fidelity for the scaled schedules are bounded by

$$p_i^{new} \ge 1 - \mathcal{F}\left(a^2\varepsilon_i^2(t)\right) = 1 - a^2\mathcal{F}\left(\varepsilon_i^2(t)\right) \tag{28}$$

We assumed $p_1 > p_2$, so $\mathcal{F}\left(\varepsilon_1^2(t)\right) < \mathcal{F}\left(\varepsilon_2^2(t)\right)$, and for any $a$ it remains true that

$$a^2\mathcal{F}\left(\varepsilon_1^2(t)\right) < a^2\mathcal{F}\left(\varepsilon_2^2(t)\right) \tag{29}$$

and therefore

$$p_1^{new} \ge p_2^{new} \tag{30}$$

We note that this holds true for many choices for $\mathcal{F}[f(t)]$, as long as

$$\mathcal{F}[af(t)] = q(a)\mathcal{F}[f(t)] \tag{31}$$

for some monotonically increasing function $q$. $\qquad\square$

## C Solving the Schrödinger equation for the adiabatic evolution

It is possibly numerically integrate and solve differential equation in Eq. 9, using the explicit evolution Hamiltonian $\mathcal{H}(s)$ for every $0 < s < 1$, and the boundary condition $|\psi(t = 0)\rangle = |E_0(s = 0)\rangle$, where $|E_0(s = 0)\rangle$ is the known ground state of the initial Hamiltonian $\mathcal{H}_b$. This first order differential equation, could be solved numerically to obtain $|\psi(t = t_f)\rangle$ in the following fashion:

1. Divide the time axis to M slices $1..M$

2. For every time slice, find $\mathcal{H}_m = \mathcal{H}(s(t = \frac{t_f}{M} \cdot m))$

3. Calculate the eigenvalue decomposition of $\mathcal{H}_m$: eigenvectors $V_i$ and eigenvalues $E_i$

4. Find the projection of the last quantum state $|\psi_{m-1}\rangle$ onto the eigenvectors space

$$|\psi_{m-1}\rangle = \sum_{i=1}^{N} a_i V_i \tag{32}$$

$$a_i = \langle V_i, \psi_{i-1}\rangle \tag{33}$$

5. Evolve the quantum state according to

$$|\psi_m\rangle = \sum_{i=1}^{N} e^{iE_i \cdot \frac{t_f}{M}} \cdot a_i \cdot V_i \tag{34}$$

6. Repeat steps 2-5 until reaching $t_f$

## D Additional Grover search results

The results of Grover search for $n = 8$ qubits, for our model, as well as the method of Lin et al. (2020) and other baselines are presented in Fig. 6.

### D.1 $n' > n$ experiments for the $n = 16$ model

To demonstrate our approach's ability to employ a model of a certain size for larger problems, we present result for sizes $n' = 17, 18, 19, 20$ using the model trained for $n = 16$.

The results are shown in Fig. 7. Our predicted schedule greatly outperforms the baseline linear schedule, with even greater advantage for larger problem sizes. We also compare to the same Hamiltonian model, trained with the MSE loss. As can be seen, this model outperforms the linear model, but is less effective than the model trained with $\mathcal{L}$.

## E Alternative training sets

The training set of the Hamiltonian model contains 10,000 random QUBO problems and 500 random Hamiltonian problems, see Sec. 3.4. Fig. 8 depicts the effect of training on the first group only, i.e., on the Hamiltonian forms for the QUBO problems. This is shown for both the 3SAT problem and the Grover problem. As can be seen, there is a relatively small drop in performance for the 3SAT problems and a signifcant one for the Grover problem. Note that in both cases, we cannot compare to the QUBO model. For the 3SAT problem, there is a polynomial overhead in size, when using the QUBO form (Glover et al., 2018). For the Grover problem, the QUBO problem is undefined.

In another set of experiments, we varied the size of the training dataset. In addition to the 10000+500 samples (of the two types mentioned above), we employed sets of size 1000+62, 2500+125, and 5000+250. Fig. 9 presents the results for 3SAT problems. Evidently, adding more training samples helps. However, there is, as expected, a diminishing returns effect. Note that the 3SAT problem is not captured by neither the random Hamiltonians nor by the random QUBO problems. Therefore, the success on these instances indicates a generalization capability.

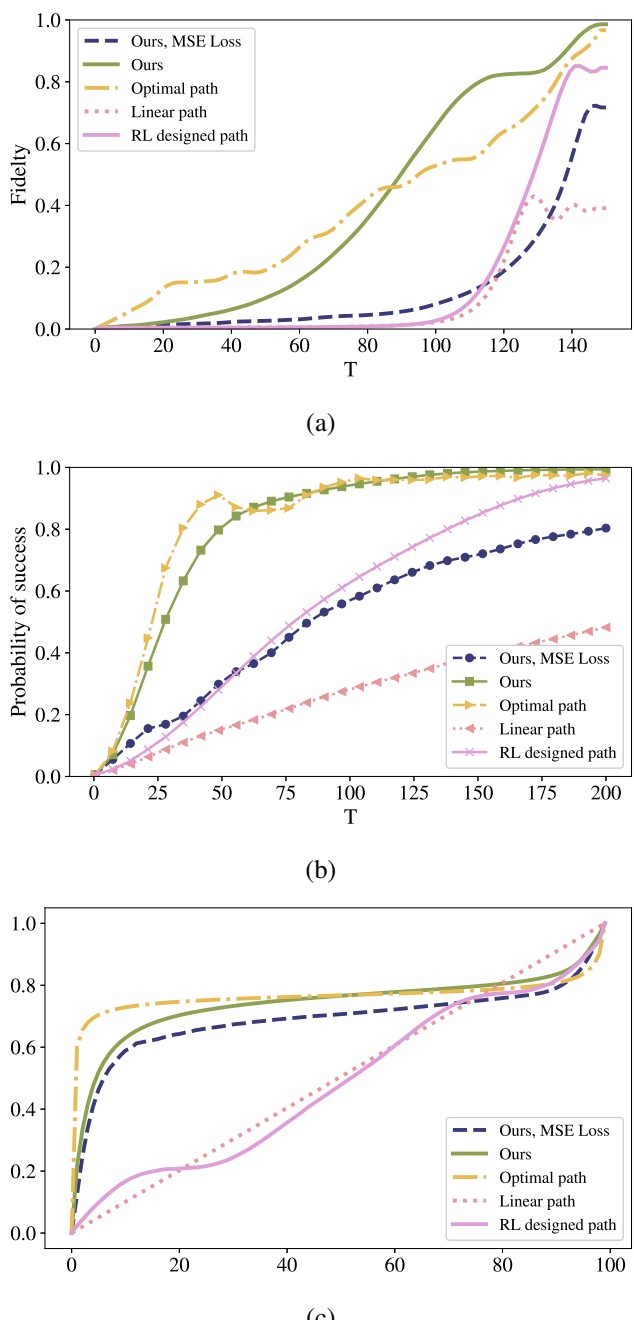

Figure 6: Model Results for Grover search over n=8 qubits. (a) fidelity for $t_f = 150$. (b) fidelity for multiple $t_f$. (c) suggested schedules for one specific instance.

# F    COMPARING THE ALTERNATIVE LOSS TERMS

In this work, a novel loss function was presented, that allowed training neural networks with better performance than standard losses. The suggested loss function is justified by our derivation in Sec. 3.3. It is further supported by all experiments conducted and for both the Hamiltonian and the QUBO networks.

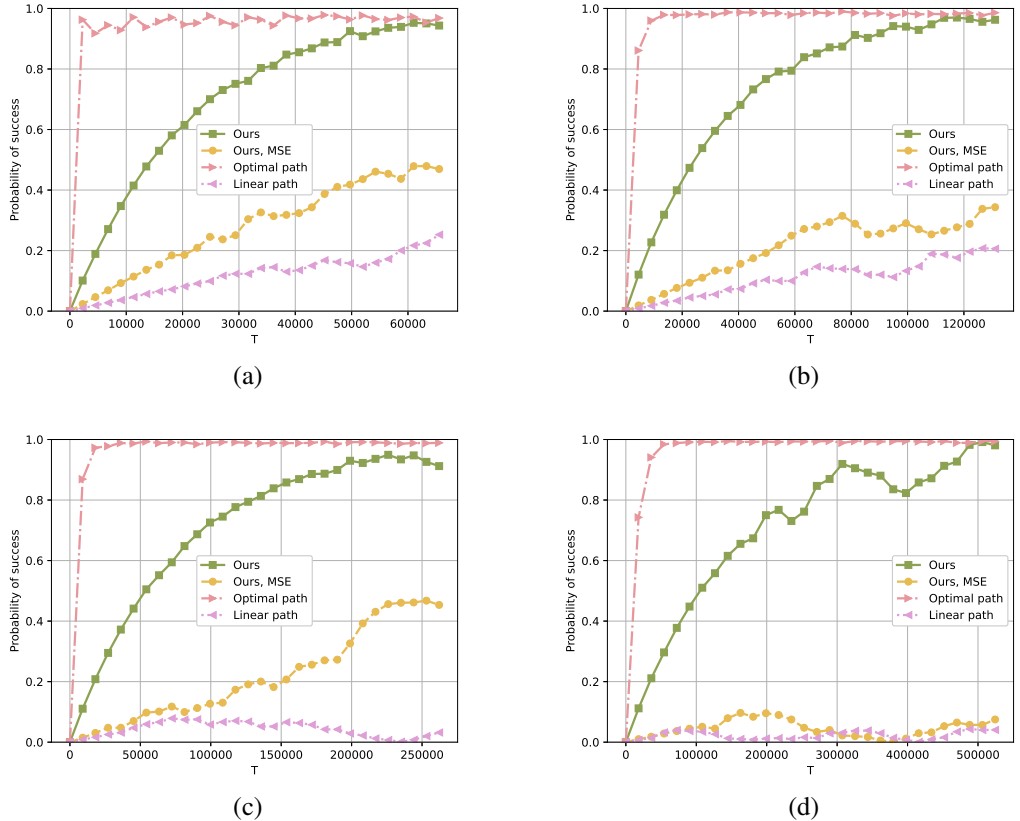

Figure 7: Fidelity for various $t_f$ values for Grover search problems of size $n'$ larger than the $n = 16$ for which the network was trained. (a) $n' = 17$, (b) $n' = 18$, (c) $n' = 19$, (d) $n' = 20$. Note the different time scale for each problem size.

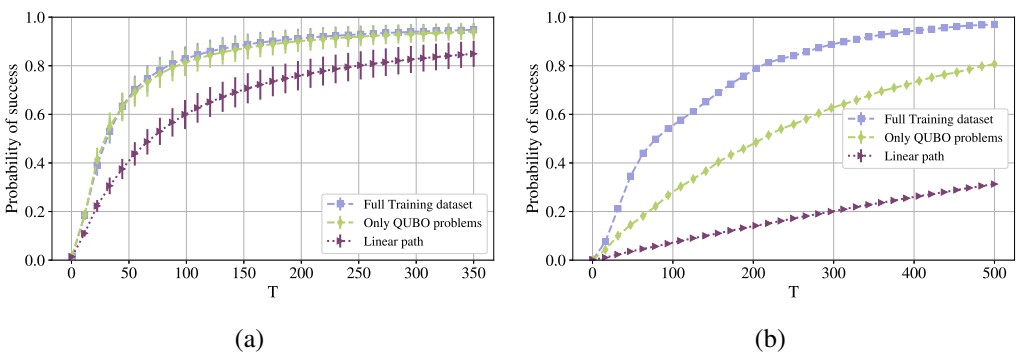

Figure 8: Success probability when training with and without the random Hamiltonians. (a) 3SAT problems of size $n = 10$. (b) The Grover problem of size $n = 10$.

To visually demonstrate the advantage of our loss function, we present a specific example. We consider for a single 3SAT problem the optimal path and three variants of it. In the first, we add random noise to $s(t)$. in the second, we shift the optimal path by a constant. The third variant adds a linear function of $t$ to it. We also consider the path that was obtained by employing $\mathcal{L}$ or MSE, see Fig. 10(a).

As can be seen in Fig. 10(b), the best path is the optimal one, followed by the path of our full method, our method with MSE, and the optimal path with the added linear factor. As can be seen in Tab. 1,

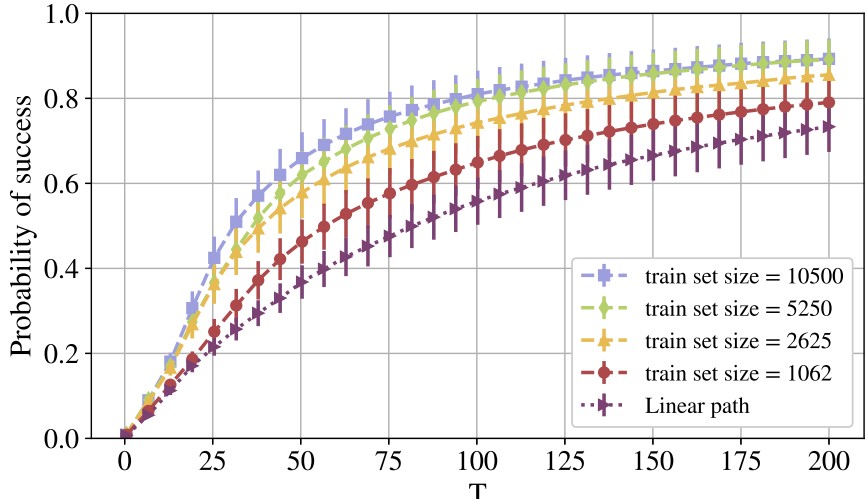

Figure 9: The effect of changing the size of the training set for 3SAT problems of size $n = 10$.

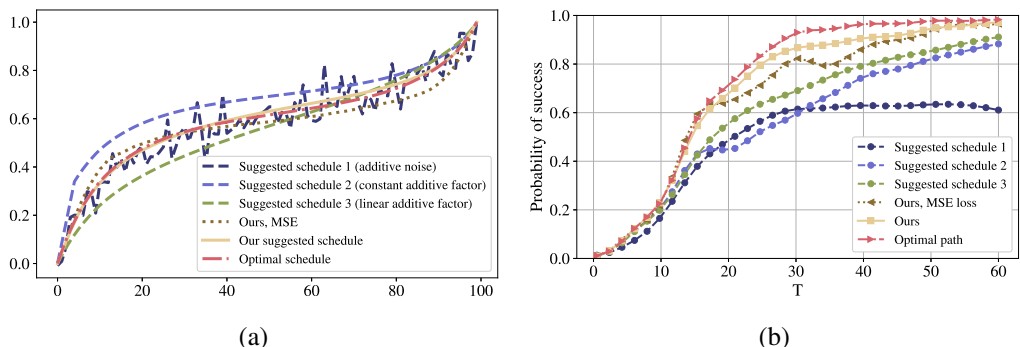

Figure 10: (a) Six different paths that were compared for a single 3SAT problem. (b) The obtained success probabilities.

our loss is predictive of the success probability, while the MSE is less so. Specifically, the MSE loss assigns a relatively low loss to the optimal path with the added Gaussian noise, while our method predicts that it would result in a low success probability.

To generalize this sample, we compare the ability of the two loss terms to identify the path that would obtain a fidelity of 0.8 faster. This discrimination ability is visualized via a receiver operating characteristic (ROC) curve.

A set of 250 test problems, randomized in a similar fashion to the training dataset of QUBO problems with $n = 8$, was generated and its optimal path $s^*$ was computed. For each problem, two possible schedules, $s_1$ and $s_2$, were randomized. Following Lin et al. (2020), who showed that the Fourier spectrum is an effective representation for paths, we sample the coefficients of the paths in the Fourier domain.

For each loss, we compute the score of the two paths with respect to the optimal path, and compute the ratio of the score associated with $s_1$ and the one associated with $s_2$. For our loss, this is given as $\frac{\mathcal{L}(d1, s^*)}{\mathcal{L}(d_2, s^*)}$, where $d_1, d_2$ are the difference vectors obtained form the paths $s_1, s_2$, respectively. We simulate both paths, and assign a label of 1 if $s_1$ leads to the probability threshold on 0.8 faster than $s_2$, 0 otherwise.

Table 1: The probability of success at a certain time point for each path presented in Fig. 10(a) and the loss obtained by both MSE and $\mathcal{L}$ in comparison to the optimal path. The high value of 1.2353 is not a typo. It is a result of our loss penalizing jittery paths.

| Path | Probability of success at T = 45 | MSE loss | Our loss |
|------|----------------------------------|----------|----------|
| Optimal with Gaussian noise | 0.63 | 0.0029 | 1.2353 |
| Optimal with a constant shift | 0.77 | 0.0070 | 0.0119 |
| Optimal with a linear shift | 0.82 | 0.0043 | 0.0099 |
| Our with the MSE loss | 0.89 | 0.0014 | 0.0061 |
| Our with $\mathcal{L}$ | 0.91 | 0.0043 | 0.0010 |
| Optimal | 0.97 | 0 | 0 |

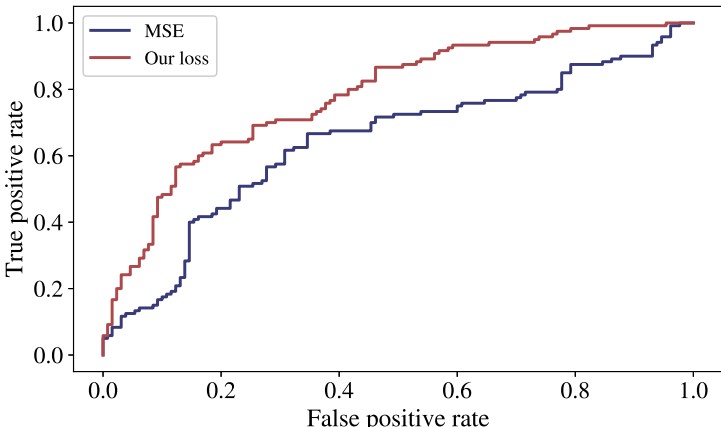

Figure 11: ROC curves for identifying the better path out of two options, see text for details. The red curve is for the $\mathcal{L}$ loss term and the blue for the MSE loss.

We compare the resulting ROC curve for $\mathcal{L}$ and for the MSE loss in Fig. 11. It is evident that the suggested loss function is more discriminative of better paths than the the MSE loss.

## G    TRAINING DYNAMICS

In Fig. 12, we present the evolution of the training and validation losses during model training. This is shown both for $\mathcal{L}$ and for the MSE loss for the Hamiltonian model of size $n = 10$.

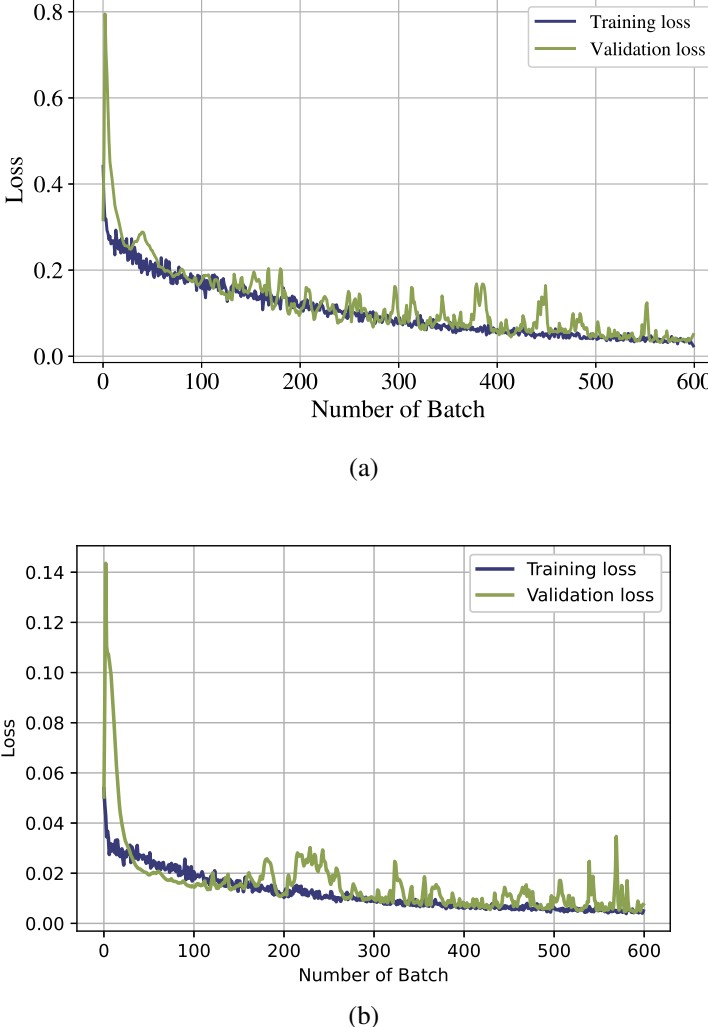

(a)

(b)

Figure 12: The loss of the training and validation sets during the training process of the $n = 10$ Hamiltonian model. (a) Training to optimize the loss term $\mathcal{L}$. (b) Optimizing the MSE loss.

