# OpenReview forum: "Fidelity-based Deep Adiabatic Scheduling"
_ICLR.cc/2021/Conference — ICLR 2021 Spotlight_

### Official Review · AnonReviewer3 · 2020-10-28
**Review of "Fidelity-based Deep Adiabatic Scheduling"**

**Rating:** 6
**Confidence:** 4

**Review:**

I would like to thank the authors for the rebuttal. The added experiments have made this work better.

While the collection of training data still seems hard (require exponential computation), I like the idea and the new experiments on generalizing from small training size to large ones. I still think the application domain of the proposed work is limited. The application to quantum computing is definitely interesting, but adiabatic scheduling is a more restricted domain and the contribution in this work is not applicable to gate-based quantum computers explored by Google, IBM, Rigetti, etc.

Because the revised manuscript is stronger now, I have modified my rating accordingly.

##########################################################################

Summary:

This paper concerns a specific application of machine learning in quantum computing: training classical machine learning models to predict the optimal schedule in adiabatic quantum computation. Adiabatic quantum computation depends greatly on the schedule to achieve good quantum speed-up. While in various cases, the optimal schedule is known. There will be instances where the optimal schedule is unknown, and the work proposes predicting those optimal schedules using neural networks. The main contribution comes from designing a new loss function for training such a model.

##########################################################################

Reasons for score:

I think the application is relatively narrow, even within the field of machine learning/quantum physics. The proposed neural architectures are very standard. The novelty comes from how to define a good loss function. The loss function is well-motivated and the empirical performances are convincing. However, the contribution may be too specific to be used in other machine learning/quantum physics applications, so I think most audiences attending ICLR may not be very interested in this result.

##########################################################################Pros:

Pros:

1. The proposed loss function is well-motivated.

2. The proposed method works well on a wide range of tasks (QUBO, Grover search, 3-SAT).


##########################################################################

Cons:

 1. The application of the contribution of this work is relatively narrow (producing schedules for adiabatic quantum computation).


2. The problem size tested by this work is small (6, 8, and 10). The QUBO problems that can be solved by current quantum devices (such as D-Wave machines) seems to be much larger (bigger than 1000).


3. The proposed machine learning approach can not be used for larger n (e.g., n > 100) because the input to the classical neural network is a 2^n vector. If this scalability issue can not be addressed, then the proposed method would not be useful in practice (any form of quantum advantage would become vacuous if we need an exponential preprocessing time to find the schedule).


##########################################################################

Questions during the rebuttal period:


1. I have some concerns about the scalability of the proposed technique as discussed in the Cons section. The input to the neural network is a 2^n dimension vector. Do the authors have ideas and supporting evidence for how one could scale to a large system when n is like 50~100?

2. Are there other non-ML alternatives to the linear schedule that people employ in practice? If the importance of scheduling is as claimed, I believe there should be extensive literature studying heuristics scheduling for QUBO and 3SAT. How does the proposed ML method compare to these existing heuristics?

---

> ### Author Response · Authors · 2020-11-25
> **Thank you for your review**
>
> Thank you for the review, we appreciate the feedback.
>
> Regarding scope: the general problem of determining an optimal evolution schedule is common and could be found in many practical fields [A]. A notable example is crystal growing (for electrooptics and semiconductors), which requires a delicate forming process which includes a phase shift. The production of high-quality industrial crystals is notoriously difficult. In general, any manufacturing process that has several controllable critical parameters (temperature, pressure, etc.), which are changed during the process to achieve some specified desired end result, is difficult to optimize with classical methods.
>
> In many of these cases, one can simulate the process for a given schedule and obtain a training set containing paths and outcomes, even if the optimal schedule is not obtainable. Then, a meaningful loss, such as the one presented in our work, could be of value for learning more effectively. The loss we study is based on the local limit of the speed of evolution. It is reasonable that such local upper bounds exist in other domains as well.
>
> Our work is consistent with the type of work presented at ICLR. We prove (Thm. 1) that a solution for a given problem can be described in a certain way. We then propose a neural network method to map the input to this output. Finally, we derive a new loss term that is tied to the physical quantity that we wish to optimize. We believe that the ICLR audience is curious about applications of ML to the emerging field of quantum computation, especially to adiabatic computation, which is where most hardware efforts are currently focused. The paper is accessible even without a background in quantum computation (R2 writes “the paper is easy to read and clearly describes basic ideas and required technical details”).
>
> Regarding scalability. We note that even though D-wave’s machine has ~1000 physical qubits, the inter-qubit connectivity is partial, and, in practice, it gets up to 64 logical qubits in a general QUBO setting [B]. We believe that the current era of quantum computing (sometimes referred to as “NISQ”, noisy intermediate-scale quantum) could greatly benefit from algorithms that allow for better performance for intermediate problem sizes.
>
> We also note that we present a model that is based on the QUBO formulation as an input. As described in the paper (Sec. 3), the QUBO formulation allows us to describe a wide variety of optimization problems and scales quadratically (n^2) with input size. It further employs convolutional layers, on the input layers.
>
> The main bottleneck for growing the size of the problems we consider is the collection of training data. The collection requires finding a few eigenvalues of a sparse symmetric matrix. Computationally one needs to multiply the Hamiltonian with a group of K vectors as part of a power iteration and to orthogonalize the group of K vectors every few iterations. The type of Hamiltonians we use for training seems to have well-behaved spectrums, leading to convergence in 10 iterations or less. Furthermore, the Hamiltonian we use for training is very sparse, with the number of elements growing as 2^{n+c}, for a small constant c, and not 2^{2n} (the sparsity is 0.02% for n=16). Therefore,  the matrix-vector multiplications can be performed efficiently.
>
> We note that our implementation was not optimized for size in any way and our experiments were performed on a single machine (no GPUs were used during data collection). The scale of the work in which we conducted experiments was mostly dictated by previous work. By using multiple machines, we can collect enough samples for training size n=22 in a relatively short amount of time. With further optimization and added parallelism, larger sizes are obtainable.
>
> Another way to address scalability is by applying models trained on smaller datasets to larger problems. Steps in this direction were already taken in the uploaded manuscript. We show that it is possible to solve a larger problem (e.g. of size 16) using a model of smaller size (10) by applying some simple manipulation to the input (pooling), see Sec. 4.1 of the revised manuscript. Using the same code, without any additional effort, we trained a model of size n=16. We further demonstrate its scaling capabilities by using it to solve larger problems (n’=17,18,19,20), see Appendix D of the revision.
>
> Following the review, we added citations for non-ML alternatives, see Sec.1 of the revised manuscript. Such approaches were not demonstrated to lead to higher fidelity and were not considered as baselines by the RL work we compare to.
>
> [A]  Jaluria, Yogesh. Advanced materials processing and manufacturing. Cham, Switzerland: Springer, 2018.
>
> [B] Okada, Shuntaro, Masayuki Ohzeki, Masayoshi Terabe, and Shinichiro Taguchi. "Improving solutions by embedding larger subproblems in a d-wave quantum annealer." Scientific reports 9, no. 1 (2019): 1-10.

---

### Official Review · AnonReviewer1 · 2020-10-28
**interesting approach to learning annealing schedules, thorough investigation of the proposed models is lacking**

**Rating:** 6
**Confidence:** 4

**Review:**

**summary**
the paper proposes to learn parametric form of optimal quantum annealing schedule. Authors construct 2 versions of neural network parameterizations mapping problem data onto an optimal schedule. They train these networks on artifically generated sets of problem of different size and test final models on the Grover search problem as well as 3SAT. Experiments demonstrate improved performance in comparison to existing approaches.

**pros**
* the approach is straightforward to implement
* experiments show the ability to do zero-shot generalization to different problem classes
* empirically proposed models perform better then baselines


**cons**
* the mechanism of observed generalization to unseen problem instances is unclear. Training dataset size is chosen to be fixed 10000, it is unclear whether the learned model isn't just memorizing the inputs but rather providing useful generalization. Experiments with varying the dataset size say as 2^m could help clarify this. In addition to that providing training curves with train and validation errors as function of step would be also very useful.
* The authors don't provide any intuition for why MSE objective performs worse then  success probability objective. Is it the case that training the latter is easier for some reason?


**questions**
* it is not clear how the architecture is varied for different model sizes. Since the input size changes with problem size, neural networks must vary at least in the input layer. Do they share the rest of the layers between different problem sizes?

**comments**
* it would be useful to add comparison b/w QUBO model and Hamiltonain model for the set of QUBO problems. Is it true that QUBO models performs better in this case?

---

> ### Author Response · Authors · 2020-11-25
> **Thank you for your review**
>
> Thank you for your review.
>
> Regarding the generalization capabilities, we would like to emphasize that our models were mostly benchmarked using problems from the 3SAT family and the Grover search, which are completely different from the training domain.  To further demonstrate this generalization capability, we present results for the exact same model (no retraining) on the MAX-CUT problem.
>
> Following the review, we added to the paper an experiment showing the performance for different sizes of the training set (Appendix E). We also added a graph showing the training and validation loss during model training, as suggested. (Appendix G).
>
> Regarding the loss function, our suggested loss is derived to approximate a real physical quantity, i.e., the success probability. Since this quantity reflects the goal of the computation, optimizing for this goal should improve the benchmark results. This is indeed evident across all experiments.
>
> In order to provide more intuition, we consider the behavior of the new loss on perturbed samples and compare it to the MSE loss (Appendix F). The qualitative example is further supported by examining the ability of each loss to identify the best path out of two options.
>
> The model architecture is mostly constant regardless of the problem size. For the Hamiltonian model, the input size is the only variation between the models for larger problem sizes. For the QUBO model, the input layer (a convolutional layer) to the model remains the same. The first fully connected layer (after the convolutional layers) scales with the problem size, and the rest of the architecture is shared for all sizes.
>
> A direct comparison between the Hamiltonian model and the QUBO model is given in what is, in the revised version, Fig. 5 (previously Fig. 3). This is done for random QUBO problems, where the QUBO model does indeed perform better than the Hamiltonian model for the relevant test cases. Such a comparison is not possible for 3SAT, since there is a polynomial overhead in the problem size when employing the QUBO representation. There is no known QUBO representation for the Grover problem, as far as we can ascertain.

---

### Official Review · AnonReviewer4 · 2020-10-29
**original and convincing work on learning interpolation schedules for adiabatic quantum computing**

**Rating:** 9
**Confidence:** 5

**Review:**

The authors consider the problem of providing optimal scheduling schemes for adiabatic quantum computing (AQC), i.e. with a way of interpolating / evolving a Hamiltonian from its initial form to its final form such that it is nwither too slow nor too rapid and thus can harness the speedup that AQC offers for solving combinatorial optimization problems (or, to be specific: QUBOs).
Indeed, optimal scheduling of is a problem of theoretical as well of practtical concern as it makes or breaks the success of AQC but general, closed form solutions are hard to come by (or simply unknown at this point in time).
Addressing drawback of previouslt proposed reinforcement learning approaches to this problem, the authors propose to train deep neural networks. To this end, they introduce a novel loss function loss that maximizes the fidelity based on an approximation of the success probability. In practical experiments, they find that schedules learned this way outperform those from previous approaches and, in addition, generalize from one type of problem to different instances of problems. Their experiments also reveal the approach to outperform previous.

This paper presents, original, convincing, and interesting work on a problem of considerable practical importance in adiabatic quantum computing. The idea of using neural networks in order to learn good (optimal) schedules for AQC is elegant and apparanetly leads to very good results. Moreover, to those with a background in AQC and deep learning, the paper is easy to read and clearly describes basic ideas and required technical details.

---

> ### Author Response · Authors · 2020-11-25
> **Thank you for your review**
>
> Thank you for your detailed and supportive review.

---

### Official Review · AnonReviewer2 · 2020-10-29
**Promising for designing quantum computers, although some potential scaling issues**

**Rating:** 8
**Confidence:** 4

**Review:**

Summary:
Deep neural networks are used for supervised learning of optimal optimization schedule for various problems in adiabatic quantum computing (AQC). The usefulness of the method is assessed on three families of optimization problems.

Strengths:
The supervised learning method is straightforward and general, yet is the first such method to be employed for adiabatic scheduling.

The method show impressive performance on all of the problem instances, outperforming both simple baselines and recently proposed methods using reinforcement learning.

In contrast to previous ML methods, the proposed method doesn't require simulations of the adiabatic evolution during training, with all such work being done upfront, in preparation of the training set.

Critiques:
In order to apply the proposed supervised learning method, the ability to compute the optimal adiabatic schedule with reasonable accuracy is already required. To ensure this method has relevance for nontrivial problems in AQC, I would request that the authors give more evidence for the generalization capabilities of their method, for example by assessing the performance of the model in one of the following settings: (1) With different training set sizes, or (2) Using only QUBO instances or only randomized Hamiltonians (as described in Section 3.4) for the training data, rather than the mixture of both which is currently employed. I understand that the performance on Grover search and 3-SAT gives some evidence for this generalization ability, but some more clear-cut evidence would be desirable here.

Using the literal Hamiltonian as an input to the supervised problem scales quite poorly with increasing problem sizes. Could the authors comment on how to deal with this scaling issue? In particular, do the authors see any means of giving an efficient parameterization for the Grover search and 3-SAT problems in lieu of the Hamiltonian diagonal elements, along the same lines as how QUBO problems can be specified via the instance matrix Q.

Can the authors comment on why the "optimal" path in Figure 1 sometime gives a worse final probability of success than the supervised learning method proposed?

Recommendation:
While the paper leaves open some questions about the generality and scalability of the proposed method, the novelty and performance of the method is promising. I would recommend acceptance.

### UPDATE AFTER THE REBUTTAL

Many thanks to the authors for revising the paper, the new material is comprehensive and does a lot to address my questions about the feasibility of scaling up the model. I would know the authors are short on space, but I would request that the current Figure 2 be tweaked in some manner to makes the legend in Figure 2c bigger. The current legend is nearly illegible, and the difference between the model trained on size 10 vs 16 is important. Independently of this, I have raised my score in light of these new additions.

---

> ### Author Response · Authors · 2020-11-25
> **Thank you for your review**
>
> Thank you for your review.
>
> Regarding the generalization capabilities, we would like to emphasize that our models were trained on randomly generated problems and were benchmarked mostly using different test problems: 3SAT and Grover search.
>
> To further demonstrate this capability, we added experiments on the MAX-CUT problem. We use the same networks as used for the benchmarks in the paper.
>
> Regarding scaling of the Hamiltonian model:
> (1) We show in Sec. 4.1 that it is possible to solve a large problem (e.g. of size 16) using a model of a smaller size n=10 previously trained by subsampling the input.
> (2) We trained a larger model (size 16) using the same approach and used it successfully to solve larger problems (sizes 17,18,19,20). The size of the biggest model trained earlier (n=11), was based on previous work. However, with our current code, we can generate a training sample for n=22 on a single machine in a day, and data collection at larger scales is entirely feasible.
>
> Regarding more efficient representations. This is indeed done in the literature for specific Hamiltonians [A]. For the computational problems benchmarked in our paper: the 3SAT problem could be represented using QUBO form, with a polynomial overhead for the number of qubits. At the sizes considered in the current work, the Hamiltonian representation is smaller than the QUBO one. Other, more efficient representations may exist. We do not know of smaller representations for the Grover problem. If such representations are found, we believe that our method, including the new loss, would be relevant, since the advantage of our method is currently evident for both the Hamiltonian and the QUBO representations.
>
> The optimal schedule in Fig. 1(b) should indeed produce a greater probability of success than other schedules. The false result was caused by insufficient sampling when generating the graph. We replaced the graph with a more accurate version. Thank you for drawing our attention to this issue.
>
> [A] Morifuji, Masato. "Size Reduction of Hamiltonian Matrix for Large-Scale Energy Band Calculations Using Plane Wave Bases." Journal of the Physical Society of Japan 87.1 (2018): 014003.

---

### Author Response · Authors · 2020-11-25
**Summary of changes**

Following the reviews, we have made these changes to the manuscript.

1. MAX-CUT experiments were added, see Sec. 4.3. These demonstrate that the learned schedule improves performance in another NP-complete domain that is far removed from the random training set.

2. We demonstrate, on the Grover search (Sec. 4.1), the ability to use a network trained for n=10 on larger values for n’=12,14,16. In addition, we train a network for n=16 and compare to it. We then use the network trained for n=16 to scale further to sizes n=17,18,19,20 (Appendix D).

3. We present, in Appendix E, (i) the results of training the Hamiltonian input network only on the QUBO problems, and (ii) the results of training on varying training set sizes.

4. To further investigate the new loss, we present a qualitative comparison between the losses. This is further illustrated by comparing the ability of each method to identify a better path out of two options (Appendix F).

5. We provide graphs containing training loss and validation loss during model training for both losses (Appendix G).

6. We fixed a numerical issue with Fig. 1(b). The probability of success for the optimal schedule now dominates that of all other schedules.

7. Citations to scheduling heuristics that do not involve machine learning were added to Sec. 1.

We are grateful to the reviewers for their constructive feedback.

---

### Decision · Program_Chairs · 2021-01-07
**Final Decision**

**Decision:**

Accept (Spotlight)

**Comment:**

All reviewers are for accepting the paper: in particular, R1 and R3 found the rebuttal sufficiently convincing to increase their scores from their initial assessment leaning towards rejection.

Strengths:
+ Clarity
+ Simplicity of the proposed approach
+ Convincing experiments outperforming reasonable baselines across all problem instances

Weaknesses:
+ Scale (as noted by R2 and R3) to larger problem sizes, beyond the setting of less than a dozen.

I agree with some hesitation that the paper is narrow in scope (both in interest from the community and scale---and ultimately whether it would interest the overall quantum computing audience). However, I think the paper makes significant advances toward the area of adiabatic quantum computation.